# Targeting FAK/PYK2 with SJP1602 for Anti-Tumor Activity in Triple-Negative Breast Cancer

Myeongjin Jeon [1], Sungpyo Hong [2], Hyoungmin Cho [1], Hanbyeol Park [2], Soo-Min Lee [1] and Soonkil Ahn [2,*]

[1] Research Center, Samjin Pharm. Co., Ltd., Seoul 07794, Republic of Korea; mjjeon@samjinpharm.co.kr (M.J.); chm6050@samjinpharm.co.kr (H.C.); soominlee@samjinpharm.co.kr (S.-M.L.)

[2] Institute for New Drug Development, Division of Life Sciences, Incheon National University, Incheon 22012, Republic of Korea; hong.sungpyo@gmail.com (S.H.); qkrgksquf100@naver.com (H.P.)

\* Correspondence: skahn@inu.ac.kr

**Abstract:** Triple-negative breast cancer (TNBC) presents significant challenges due to its aggressive nature and limited treatment options. Focal adhesion kinase (FAK) has emerged as a critical factor promoting tumor growth and metastasis in TNBC. Despite encouraging results from preclinical and early clinical trials with various FAK inhibitors, none have yet achieved clinical success in TNBC treatment. This study investigates the therapeutic potential of a novel dual inhibitor of FAK and PYK2, named SJP1602, for TNBC. In vitro experiments demonstrate that SJP1602 effectively inhibits FAK and PYK2 activities, showing potent effects on both kinases. SJP1602 shows concentration-dependent inhibition of cell growth, migration, invasion, and 3D spheroid formation in TNBC cell lines, surpassing the efficacy of other FAK inhibitors. Pharmacokinetic studies in rats indicate favorable bioavailability and sustained plasma concentrations of SJP1602, supporting its potential as a therapeutic agent. Furthermore, in TNBC xenograft models, SJP1602 exhibits significant dose-dependent inhibition of tumor growth. These promising results emphasize the potential of SJP1602 as a potent dual inhibitor of FAK and PYK2, deserving further investigation in clinical trials for TNBC treatment.

**Keywords:** triple-negative breast cancer; FAK inhibitor; targeted cancer therapy; metastasis suppressor; molecular modeling





## 1. Introduction

Triple-negative breast cancer (TNBC) is a subtype of breast cancer that lacks targeted treatments and accounts for 15–20% of incident breast cancers [1]. TNBC is characterized by the absence of human epidermal growth factor receptor 2 (HER2) expression, as well as minimal expression of estrogen receptors (ERs) and progesterone receptors (PRs) based on clinical assays [2]. This subtype is known for its aggressive nature, with cancer cells exhibiting a high-grade phenotype and rapid proliferation. These aggressive cellular characteristics contribute significantly to the unfavorable prognosis associated with TNBC compared to other breast cancer types. The limited treatment options available for TNBC exacerbate the challenges in managing this disease [1,3].

Focal adhesion kinase (FAK) is a cytoplasmic protein tyrosine kinase that is typically responsible for transducing signals from cellular adhesions. It governs a wide array of cellular biological functions, notably the survival, migration, and invasion capabilities of cancer cells [4,5]. The interaction of integrin receptors with the extracellular matrix (ECM) leads to the recruitment of FAK to areas of integrin clustering, commonly referred to as 'focal adhesions' [6]. Upon localization to focal adhesions, FAK undergoes a conformational change, enabling autophosphorylation of the tyrosine (Tyr,Y) residue 397. Following this, the phosphorylated Y397 functions as a binding site for the SRC family kinases containing SRC homology 2 [7]. This event culminates in the creation of an active FAK-SRC signaling complex capable of initiating many downstream signaling pathways [8].

FAK is central in numerous signaling pathways that drive cancer proliferation and metastasis [5]. FAK survival signals, triggered by integrins and other extracellular stimuli, play a crucial role in hindering anoikis and various types of cell death [9]. In simplistic terms, the enhancement of survival signals due to increased FAK activity in vitro parallels with enhanced tumor growth. In human breast cancer cell lines, FAK depletion has been demonstrated to obstruct tumor growth, which was primarily induced by oncogenic mutations in the PI3K and RAS signaling pathways [10]. Essentially, these studies emphasize the role of FAK as a key regulator of intrinsic cellular signals that promote proliferation. Canonical FAK signaling is also associated with the formation and disassembly of focal adhesions [11]. FAK plays a key role in dynamically reorganizing the actin cytoskeleton, which is a fundamental aspect of cellular motility and protrusion [5]. FAK activity is linked to the upregulation of MMP9 expression and spontaneous metastasis in syngeneic and orthotopic mouse models of breast carcinoma [12]. FAK is the protein product of the *PTK2* gene. Contrary to typical oncogenes such as PI3K or RAS, only a limited number of missense mutations within PTK2 have been identified in tumors [13]. While mutations in the PTK2 gene are uncommon, amplification of the gene and increased FAK expression are frequently observed in a variety of cancers [4,5]. Data from The Cancer Genome Atlas illustrate that FAK mRNA levels are elevated in about 26% of invasive breast cancers, a situation that is strongly associated with reduced overall patient survival rates [14].

Proline-rich tyrosine kinase 2 (PYK2), which shares 45% of its amino acid sequence with its orthologue FAK, has been found to replace certain FAK functions in knockout mouse models after FAK loss. Additionally, in fibroblasts where FAK has been knocked out, elevated levels of PYK2 activate an intrinsic survival mechanism, further promoting cellular survival [15–17]. Given these roles, the activity of PYK2 has the potential to negatively affect the outcomes of therapeutic strategies aimed at targeting FAK.

FAK inhibitors, which primarily consist of small molecules, exert their therapeutic action by hindering the kinase-dependent activity of FAK. These molecules predominantly function as ATP competitive inhibitors, blocking the ATP-binding site to inhibit FAK kinase catalytic activity, or as allosteric inhibitors, inducing conformational changes that suppress kinase activity [18]. BI-853520 is a highly selective FAK inhibitor that demonstrates considerable potency, effectively reducing FAK autophosphorylation in preclinical models of prostate cancer. This molecule has shown potential in hindering primary tumor growth and metastasis, leading to its current evaluation in a Phase I clinical trial aimed at assessing safety and tolerability and defining the maximum tolerated dose [19]. GSK2256098 distinguishes itself by its highly selective, ATP-competitive FAK inhibitory activity. Currently in Phase I clinical trials, it has exhibited potential in reducing cell viability and growth, as well as inhibiting FAK-mediated AKT and ERK activation [20]. VS-4718 and VS-6062, potent FAK or dual FAK/PYK2 inhibitors, respectively, have demonstrated potential in preclinical models, limiting tumor progression and inhibiting tumor growth. They are now under investigation in Phase I clinical trials [21,22]. Finally, Defactinib (VS-6063), a potent dual FAK/PYK2 inhibitor, has successfully transitioned to Phase I and II trials after demonstrating promising preclinical results such as reducing tumor weight and enhancing chemosensitivity [23,24]. While all these inhibitors have demonstrated potential in preclinical studies and early phase clinical trials, further rigorous evaluations are crucial in ongoing and future clinical trials to conclusively establish their therapeutic efficacy.

However, despite the progress made in the development of FAK inhibitors, none have yet achieved clinical success in the treatment of triple-negative breast cancer (TNBC). It is noteworthy that FAK is frequently found to be upregulated in TNBC compared to normal breast tissue, suggesting the potential therapeutic value of FAK inhibitors in this specific subgroup [25]. The high expression of FAK in breast tumors holds significant prognostic value as it has been associated with more aggressive tumor characteristics, including invasion and the triple-negative phenotype. This correlation reinforces the necessity to explore FAK inhibitors as a therapeutic strategy, especially for treating more aggressive and difficult-to-manage subtypes of breast cancer, such as TNBC [26].

This study explores the potential of SJP1602, a novel dual inhibitor of FAK and PYK2, as a promising therapeutic option for TNBC. Current therapies for TNBC are inadequate, necessitating the pursuit of new, effective therapeutics. SJP1602 exhibits potent inhibitory activity against both FAK and PYK2, outperforming existing inhibitors such as VS-6063 in in vitro studies with TNBC cell lines. Notably, SJP1602 demonstrates superior inhibitory performance in 3D cultures, representing the tumor microenvironment. It also hinders migration and invasion of TNBC cells, indicating its potential in suppressing metastatic progression. In vivo pharmacokinetic studies in Sprague Dawley (SD) rats confirm the high bioavailability and sustained plasma concentrations of SJP1602, supporting its therapeutic promise. In xenograft models of TNBC, SJP1602 administration leads to significant and dose-dependent reductions in tumor growth across various TNBC cell lines.

In conclusion, the research presented in this study highlights the potential of SJP1602 as a potent dual inhibitor of FAK and PYK2, offering promising implications for TNBC treatment. These findings contribute to the advancing field of targeted therapeutics in oncology, positioning SJP1602 as a promising candidate for further clinical evaluation.

## 2. Materials and Methods

### 2.1. Materials

SJP1602 (99% purity) was synthesized by Samjin Pharmaceutical Co. (Seoul, Republic of Korea). For in vitro studies, VS-6063 was sourced from Selleckchem (Houston, TX, USA) and for in vivo experiments, it was obtained from DAEJUNG Chemicals & Metals (Seoul, Republic of Korea). GSK2256098 was procured from MedKoo Biosciences (Morrisville, NC, USA). The cell culture media, HEPES buffer solution, 1X DPBS, and antibiotics were purchased from Gibco (Waltham, MA, USA). Fetal bovine serum (FBS) was acquired from Hyclone (Logan, MA, USA). Trypsin-EDTA solution and Dimethyl sulfoxide (DMSO) were both purchased from Sigma, with the latter specifically from Sigma-Aldrich (St. Louis, MO, USA). The culture dishes (100 mm), 96-well clear round-bottom ultra-low attachment microplates, 96-well white flat-bottom microplates, and Matrigel growth factor-reduced, phenol red-free were sourced from Corning (Corning, NY, USA). Lastly, the 3D CellTiter-Glo assay kit was bought from Promega (Madison, WI, USA).

### 2.2. Modeling of FAK/SJP1602 Complex

The X-ray crystal structure of the FAK kinase domain (PDB ID: 2JKK) was downloaded from the Protein Data Bank in PDB format. For the molecular docking analysis of SJP1602 with the FAK kinase domain (PDB ID: 2JKK), Discovery Studio® 2020 (BIOVIA; San Diego, CA, USA) was utilized. In the docking studies, the structure of SJP1602 was superimposed onto the FAK kinase domain. The initial complex was optimized using 1000 steps of steepest descent and 3000 steps of conjugate gradient while restraining the FAK heavy atoms to their initial positions by means of a harmonic force constant of 1 kcal·mol$^{-1}$·Å$^{-2}$.

### 2.3. In Vitro FAK, PYK2, and Insulin Receptor (InsR) Activity

The kinase assays were performed and quantified according to the manufacturer's instructions using the ADP-Glo™ Kinase Assay Kit for FAK (Promega, V1971), PYK2 (Promega, V4083) and InsR (Promega, V3901). Each kinase reaction mixture was prepared to contain the compounds of interest. After thorough pipetting, 25 μL of the mixture was added to each well and incubated for 1 h at room temperature. Subsequently, 25 μL of ADP-Glo reagent was added, followed by a 40 m incubation at room temperature. Next, 50 μL of the kinase detection reagent was added, and the mixture was incubated for an additional 30 m at room temperature. Luminescence was measured using the Tecan Spark 10M (Integration time: 1 s) (Männedorf, Switzerland).

The kinase profiling analysis was performed at Eurofins (Poitiers, France). A total of 110 kinase assays were carried out according to Eurofins' standard protocols. For protein kinase assays, the kinase activity was measured using Eurofins' KinaseProfiler radiometric assay. In this assay, a panel of kinases was incubated with the test compound SJP1602

at a concentration of 1 μM, and the kinase activity was determined using radiolabeled phosphate as a substrate. Lipid and atypical kinase assays were conducted using Eurofins' homogeneous time-resolved fluorescence assay. The same panel of kinases was incubated with SJP1602 at a concentration of 1 μM, and the activity of lipid and atypical kinases was measured using this assay.

### 2.4. Cell Culture

The breast cancer cell lines used were MDA-MB-231, MDA-MB-453, HCC70, BT-20, and Hs578T cells, which were purchased from the Korean Cell Line Bank (Seoul, Republic of Korea), and BT-549 cells, which were purchased from ATCC (Manassas, VA, USA). The cells were grown in a humidified atmosphere with 5% $CO_2$ at 37 °C in either Dulbecco's Modified Eagle Medium (DMEM) or RPMI Medium 1640 (RPMI), supplemented with 10% FBS and penicillin.

### 2.5. FAK [pY397] Enzyme-Linked Immunosorbent Assay

TNBC cells were seeded into 60 mm dishes at a density of $1 \times 10^6$ cells/dish in 5 mL of culture media. After 24 h, cells were treated with concentrations ranging from 1 nM to 10 μM of SJP1602 for 1 h, and cell lysates were prepared to detect phospho-FAK levels. The FAK [pY397] enzyme-linked immunosorbent assay was performed according to the manufacturer's instructions. The absorbance at 450 nm was measured to quantify the results.

### 2.6. Colony Formation Assay

MDA-MB-231 cells were seeded at $2.5 \times 10^2$ cells/well in 12-well plates. FAK inhibitors (SJP1602, VS-6063, GSK2256098) were added 24 h post seeding without changing the media. After 7 days, the cells were replaced with fresh media containing 10% FBS and the same initial concentration of the FAK inhibitor. After 11 days, the cells were fixed with 4% PFA for 10 m at room temperature and stained with 0.5% crystal violet dissolved in 25% methanol for 20 min at room temperature. The cells were then washed with distilled water, dried, and images of the entire plate were obtained.

### 2.7. Two-Dimensional Cell Invasion Assay

Cell invasion capacity was analyzed using the Boyden chamber assay. The 24-well Boyden chambers with Matrigel-coated filters (8 μm pore size) were obtained from Becton-Dickinson (San Diego, CA, USA). Cells treated with vehicle or FAK inhibitors (SJP1602, VS-6063, GSK2256098) at a concentration of 5 μM were resuspended in serum-free media and $5 \times 10^5$ cells were added to the Matrigel-coated upper compartment of the invasion chambers. Fresh culture medium with 5% FBS was added to the lower compartment. After 48 h, cells on the upper side of the filter were removed using cotton swabs. The underside of the filter was fixed in 100% methanol, washed in PBS, and stained using crystal violet. Cells that invaded the Matrigel were analyzed using a microscope (Nikon, TE2000). Image analysis was performed using the color analysis function in Adobe Photoshop 2021 (Version 22.4.2, Adobe, Adobe (San Jose, CA, USA)). Quantitative data were obtained from color measurement records in Adobe Photoshop 2021.

### 2.8. Cell Migration Assay

Cell migration capacity was analyzed using the transwell migration assay. The 24-well chambers (8 μm pore size) were obtained from Corning (Corning, NY, USA). Cells treated with vehicle or FAK inhibitors (SJP1602, VS-6063, GSK2256098) at a concentration of 5 μM were resuspended in serum-free media and $5 \times 10^4$ cells were added to the upper compartment of the transwell chambers. Fresh culture medium with 5% FBS was added to the lower compartment. After 16 h, cells on the upper side of the filter were removed using cotton swabs. The underside of the filter was fixed in 100% methanol, washed in PBS, and stained using crystal violet. Migrated cells were analyzed using a microscope

(Nikon, TE2000). Image analysis was performed using the color analysis function in Adobe Photoshop 2021 (Version 22.4.2, Adobe (San Jose, CA, USA)). Quantitative data were obtained from color measurement records in Adobe Photoshop 2021.

### 2.9. Three-Dimensional Spheroid Assay

Breast cancer cell lines were cultured in medium with or without 2% Matrigel (Corning, NY, USA) for 3 days in each 96-well ultra-low attachment round plate to form a single spheroid at a density of $3 \times 10^3$ cells/well. SJP1602 was dissolved in DMSO and treated for 4 days at concentrations ranging from 200 nM to 20 μM. Cell viability was quantified according to the manufacturer's instructions using a CellTiter-Glo 3D cell viability assay (Promega, Madison, WI, USA). Luminescence was measured using the Tecan Spark 10M.

### 2.10. Three-Dimensional Cell Invasion Assay

Cells were added to an ultra-low 96-well round-bottom plate at a density of $5 \times 10^3$ cells/well and cultured for 3 days to form spheroids. The medium was removed, and a mixture of neutralized Type I Collagen and Matrigel was added to the wells. After 1 h of incubation, compound concentrations (2×) in 100 μL of solution were added per well. The plate was then incubated for 72 h. Relative invasion was quantified based on the following formula using the microscope and imaging software ImageJ (Version 2.1.0/1.54e, National Institutes of Health, Bethesda, MD, USA):

$$\text{Relative invasion} = (\text{total area of compound treated group} - \text{spheroid area of compound treated group}/(\text{total area of control} - \text{spheroid area of control}) \tag{1}$$

### 2.11. Animals

Six-week-old female BALB/c nude mice were procured from Orient Bio (Seongnam, Republic of Korea). The mice were individually housed in a controlled environment with a temperature of $22 \pm 3$ °C, relative humidity of $50 \pm 20\%$, and a 12 h light–dark cycle. The mice were provided sterilized feed and distilled water. All animal handling and experimental procedures complied with the relevant Standard Operating Procedures (SOPs). Ethics Approval: All animal experiments were carried out in accordance with the ethical guidelines and approved by the Institutional Animal Care and Use Committee (IACUC) at the following institutions:

(1) Samjin Pharmaceutical Co., Seoul, Republic of Korea (Approval Code: SJ-2018-004). Approval Period: 1 December 2018–30 November 2019.
(2) Daegu-Gyeongbuk Medical Innovation Foundation (DGMIF), Daegu, Republic of Korea (Approval Code: DGMIF-18041702-01). Approval Period: Starting from 20 January 2015.
(3) Chaon, Seongnam, Republic of Korea (Approval Code: CE2019094). Approval Period: 3 October 2019–31 January 2020.

### 2.12. Pharmacokinetics

The pharmacokinetics (PK) study was conducted in male SD rats. Three rats received a single-dose intravenous injection of SJP1602 at 5 mg/kg, and another set of three rats received a single-dose oral administration of SJP1602 at 20 mg/kg. The vehicle composition used was DMA 3% and Cremophor ELP 4%. Blood drug concentrations over time were determined by drawing blood after 0.17, 0.5, 1, 2, 4, 6, and 24 h. Plasma was isolated using K3EDTA anticoagulation tubes, and the drug concentration in the plasma was analyzed by LC/MS/MS (Waters, MA, USA). To analyze the concentration of pretreated rat plasma samples, a Waters Acquity ultra-performance liquid chromatography system (UPLC, Waters) was employed, in tandem with a C18 column (Acquity UPLC® BEH C18 1.7 μm, $2.1 \times 50$ mm). The mass spectrometry analysis was conducted using the Waters Xevo-TQ-S micro. For the mobile phases, phase A consisted of distilled water with 0.1%

formic acid, while phase B comprised methanol, also with 0.1% formic acid. PK parameters were subsequently derived using WinNonlin software (version 2.1), which enabled the determination of AUC (ng×h/mL), Cmax (ng/mL), and bioavailability (BA). The area under the drug concentration curve (AUC) was calculated utilizing the trapezoidal rule, focusing on the time interval $AUC_{0-24h}$.

### 2.13. Xenograft Mouse Model

Triple-negative breast cancer cell lines derived from humans, specifically MDA-MB-231 ($1 \times 10^6$/mouse), BT-549 ($1 \times 10^7$/mouse), and MDA-MB-453 ($2 \times 10^7$/mouse), were heterologously transplanted subcutaneously into the tissue of 6-week-old female BALB/c nude mice to initiate tumor growth. Upon establishment of tumors, with an average tumor volume reaching approximately 100 mm$^3$, mice were grouped and subjected to daily oral dosing. Treatments included: vehicle control, low concentration of SJP1602 (20 mg/kg), medium concentration of SJP1602 (40 mg/kg), high concentration of SJP1602 (80 mg/kg), and a reference compound, Defactinib (80 mg/kg). The vehicle composition used was DMSO:PEG 400:tween 80:DW = 8%:50%:10%:32%. The duration of drug administration was defined for 2 weeks based on the growth rate of the vehicle group of each xenografted tumor. The tumor volume was measured three times a week for potency measurement, and the body weight of mice was also measured three times a week for toxicity evaluation in vivo. Tumor volume was calculated based on Equation (2) and the tumor growth inhibition rate (% TGI) was calculated using Equation (3).

$$\text{Tumor volume (mm}^3) = [(\text{tumor short diameter}^2 \times \text{tumor long diameter}/2)] \quad (2)$$

$$\%\text{TGI} = 100 \times [1 - (\text{TV (Tumor volume) final treated} - \text{TVinitial treated})/(\text{TVfinal control} - \text{TVinitial control})] \quad (3)$$

### 2.14. Western Blotting

TNBC cells were seeded into ultra-low attachment microplates and incubated at 37 °C under a 5% $CO_2$ atmosphere for 72 h. After the incubation period, the cells were treated with the compound SJP1602 at its $IC_{50}$ concentration. Following this treatment, cell lysates were prepared, and the protein levels of FAK, p-FAK (Tyr397), PYK2, and p-PYK2 (Y402) were analyzed. Equal amounts of these proteins were denatured by boiling in Laemmli sample buffer for 5 min, followed by electrophoresis on sodium dodecyl sulfate-polyacrylamide gel electrophoresis (SDS-PAGE) gels. The separated proteins were then electrotransferred onto nitrocellulose (NC) membranes. To prevent non-specific binding, the membranes were blocked using 5% skim milk in Tris-buffered saline containing 0.01% Tween-20 (TBS/T) for one hour. After rinsing the blots three times with TBS/T, they were incubated overnight at 4 °C with primary antibodies: Anti-FAK (Cell Signaling, Danvers, MA, USA #3285S), p-FAK (Cell Signaling, Danvers, MA, USA #3283S), PYK2 (Abcam, Cambridge, UK #ab32571), p-PYK2 (Abcam, Cambridge, UK #ab4800), and β-actin (Sigma Aldrich, St. Louis, MO, USA #A5316), all prepared in TBS/T. The blots underwent another set of three washes in TBS/T and were then exposed to secondary antibodies: Anti-rabbit HRP-conjugated (Invitrogen, Carlsbad, CA, USA #G21234) and Anti-mouse HRP-conjugated (Invitrogen, Carlsbad, CA, USA #G21040) at room temperature for an hour. After three additional washes in TBS/T, the protein bands were detected using the ECL™ Western blotting reagent (GE Healthcare, Chicago, IL, USA #RPN2106).

## 3. Results

### 3.1. Discovery of FAK Inhibitor SJP1602

SJP1602, with the chemical formula 2-((2-((4-(4-((2s,5r)-5-hydroxyadamantan-2-yl) piperazin-1-yl)-2-methoxyphenyl)amino)-5-(trifluoromethyl)pyrimidin-4-yl)amino)-N,3-dimethylbenzamide, is a synthetically designed FAK inhibitor (Figure 1A) [27]. It functions by binding to the ATP-binding site on the FAK protein. The visualization of this interaction

is provided in Figure 1B, which presents a docking model of SJP1602, mapped against the released X-ray crystal structure of FAK (PDB 2JKK). Evaluating the binding energies from the docking analysis, we identified two key residues in the FAK protein that engage in hydrogen bonding interactions with SJP1602. Specifically, a hydrogen bond forms between the amino group of SJP1602 and the ketone group of Cys502 in FAK. In addition, an intermolecular hydrogen bond was noted between the hydroxy group of adamantane in SJP1602 and Glu506. SJP1602 also displays intramolecular hydrogen bonding, which is observed between its trifluoromethyl group and the carbonyl group of the amide, thereby enhancing ligand stability.

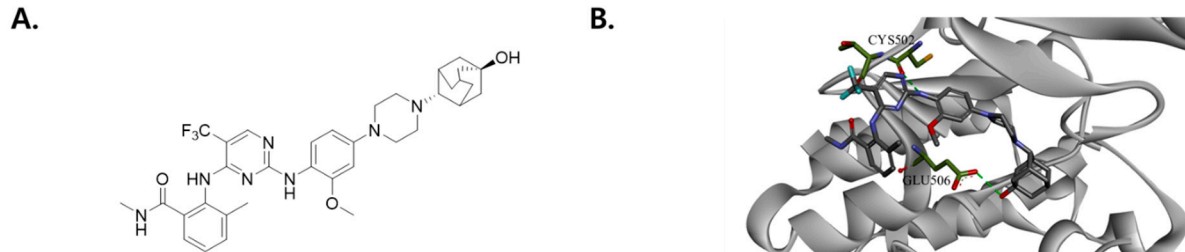

**Figure 1.** Chemical structure and binding mode of SJP1602 in FAK. (**A**) Chemical structure of SJP1602. (**B**) View of the binding mode of SJP1602 in FAK. SJP1602 and protein residues are represented as sticks with the following atom colors: carbon (SJP1602), dim gray; carbon (FAK), green; oxygen, red; nitrogen, blue; sulfur, yellow; fluorine, cyan. The green lines with dashes indicate hydrogen bonding interactions. Molecular modeling was conducted in Discovery Studio using CHARM.

### 3.2. Characteristics of SJP1602 and Its Selective Inhibition of FAK

Using an enzymatically active FAK protein, we initiated our evaluation of the inhibitory properties of SJP1602 against FAK activity in vitro. In this assay, SJP1602 showed inhibition of catalytic activity, presenting an $IC_{50}$ (half maximal inhibitory concentration) value of 3.7 nM (Table 1). In a comparable experiment, SJP1602 was observed to inhibit PYK2, showing an $IC_{50}$ of 24.7 nM. The $IC_{50}$ value for SJP1602 on Insulin Receptor (InsR) was 317 times greater than the $IC_{50}$ for FAK. SJP1602 was tested against a commercially available panel of recombinant enzymes at a concentration of 1 μM, as detailed in Supplementary Table S1. In this assay, 1 μM SJP1602, which is 270-fold the $IC_{50}$ for FAK, inhibited MET and ALK by 79% and 72%, respectively.

**Table 1.** Biochemical $IC_{50}$ of SJP1602 to FAK, PYK2, and Insulin Receptor activity. Data are presented as the mean $IC_{50}$ value from the kinase assay ± SEM (n = 3).

| Compound | Kinase ($IC_{50}$, nM) | | |
|---|---|---|---|
| | **FAK** | **PYK2** | **InsR** |
| SJP1602 | 3.7 ± 0.007 | 24.7 ± 0.919 | 1175 ± 60.1 |

In order to assess the potential of SJP1602 in inhibiting endogenous FAK activity within cells, we conducted an analysis of the phosphorylation of FAK Tyr397 across six cultured triple-negative breast cancer cell lines. FAK activity was analyzed using an enzyme-linked immunosorbent assay (ELISA) that specifically targets FAK pTyr397. The results showed that SJP1602 inhibited FAK phosphorylation in MDA-MB-231, MDA-MB-453, and HCC70 cells, presenting $IC_{50}$ values of 20, 43, and 83 nM, respectively (Table 2). These findings indicate that SJP1602 is highly effective in inhibiting FAK activity within TNBC cells. Western blot assays were conducted to detect phospho-FAK levels, and they corroborated the results obtained from the ELISA assay (Supplementary Figure S1).

**Table 2.** Cell-based IC$_{50}$ of SJP1602 on FAK kinase activity within triple-negative breast cancer cell lines. FAK activity was analyzed using an enzyme-linked immunosorbent assay (ELISA) targeting FAK [pY397]. Data are presented as the mean IC$_{50}$ value from ELISA $\pm$ SEM (n = 3).

| Compound | MDA-MB-231 | MDA-MB-453 | HCC70 |
|---|---|---|---|
| SJP1602 | 20 $\pm$ 7 | 43 $\pm$ 16 | 83 $\pm$ 17 |

### 3.3. Inhibition of Tumor Cell Growth and Invasion by SJP1602

In order to ascertain the influence of SJP1602 on the growth of TNBC cells, we executed a colony formation analysis in both the presence and absence of SJP1602. The resulting data revealed a dose-dependent decrease in the formation of MDA-MB-231 colonies within dishes exposed to SJP1602, thereby highlighting the inhibitory effect of SJP1602 on cell growth (Figure 2A,B). In contrast, SJP1602 exhibited no cytotoxicity against normal cell lines such as CCD-18Co, WI38, Fa2N4, FDF, and HEK293T (Supplementary Table S2).

**A.**

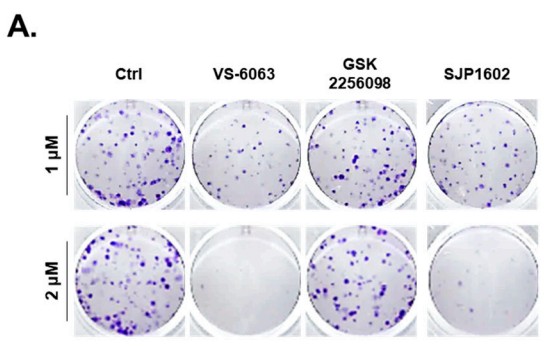

**B.**

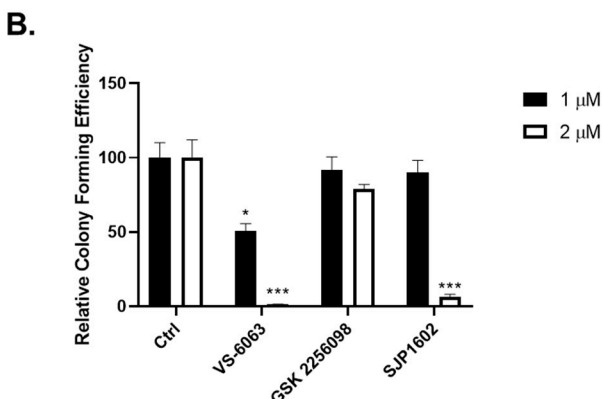

**Figure 2.** Inhibition of tumor cell growth by SJP1602. (**A**) MDA-MB-231 cells were seeded and treated with FAK inhibitors (SJP1602, VS-6063, GSK2256098) at concentrations of 1 μM or 2 μM. After 11 days, the cells were fixed and stained with crystal violet. (**B**) Bar diagram representing the colonies stained with crystal violet. Statistical analysis was conducted using an unpaired, two-tailed Student's *t*-test. * $p < 0.05$, *** $p < 0.001$ indicates a significant difference in colony forming efficiency compared to the control group.

FAK inhibitors demonstrate enhanced efficacy in 3D (three-dimensional) cultures due to the increased relevance of FAK signaling for cellular survival in these environments [20,21,28]. In 2D (two-dimensional) cultures, cells have the opportunity to spread out and adhere to the surface of the dish, maximizing their interactive surface area. Consequently, their survival is less dependent on FAK signaling. On the other hand, within 3D cultures, cells are confined by the presence of other cells and the extracellular matrix, which impedes their capacity to extend and interact as freely as in 2D cultures. As a result, their survival becomes more dependent on FAK signaling [4,29]. To evaluate the effect of SJP1602 within a 3D environment, we cultured TNBC cells in 3D Matrigel and compared the efficacy of SJP1602 with that of VS-6063 and GSK2256098. The TNBC cells, grown as 3D spheroids, were treated with increasing concentrations of SJP1602 (0.2–20 μM) over a four-day period to monitor its effects on spheroid growth (Figure 3). Notably, these TNBC cell lines, known for their elevated FAK expression and activity that promote proliferation and invasion [30], responded to SJP1602 with a concentration-dependent inhibition of cell growth.

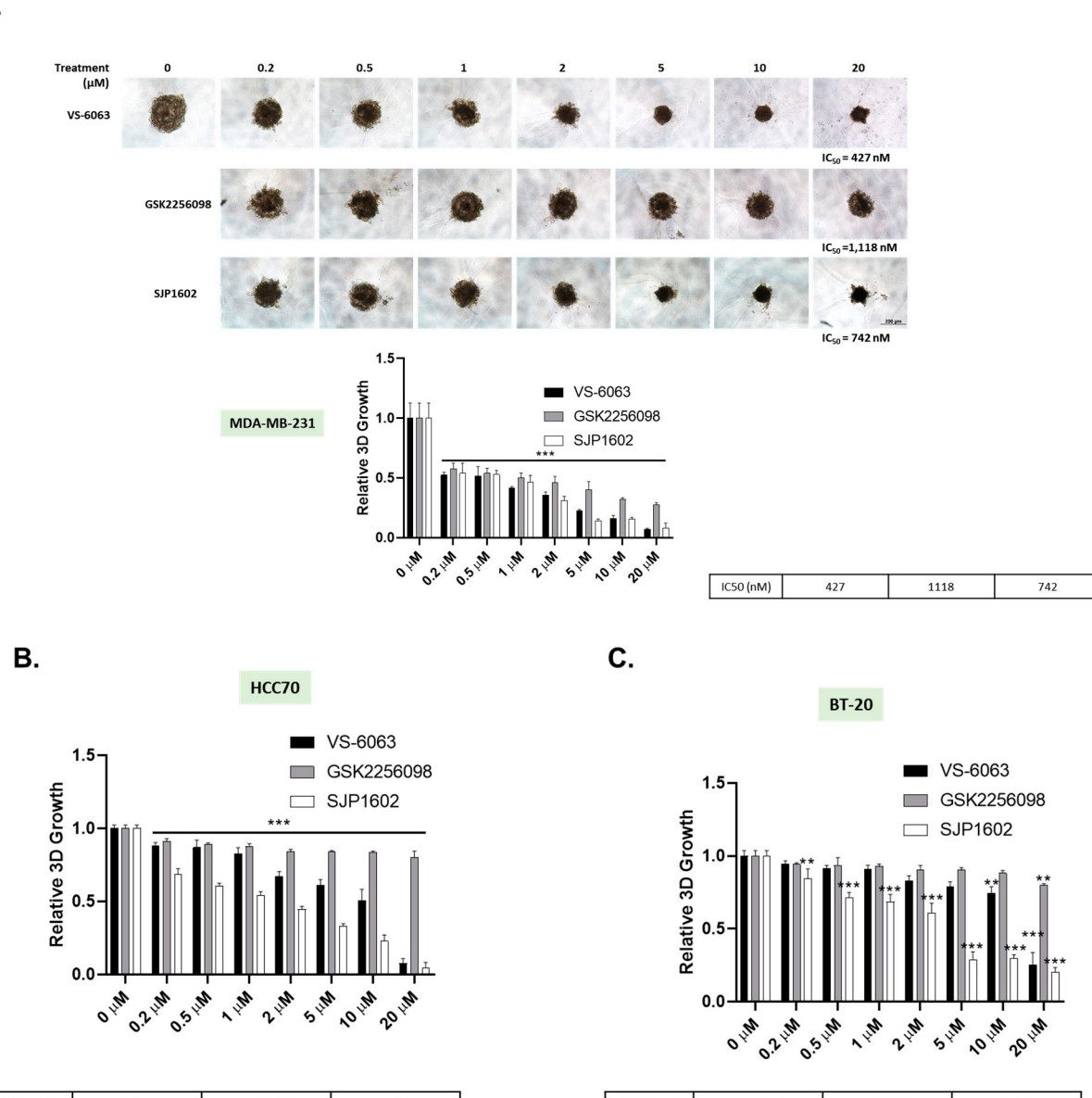

**Figure 3.** *Cont.*

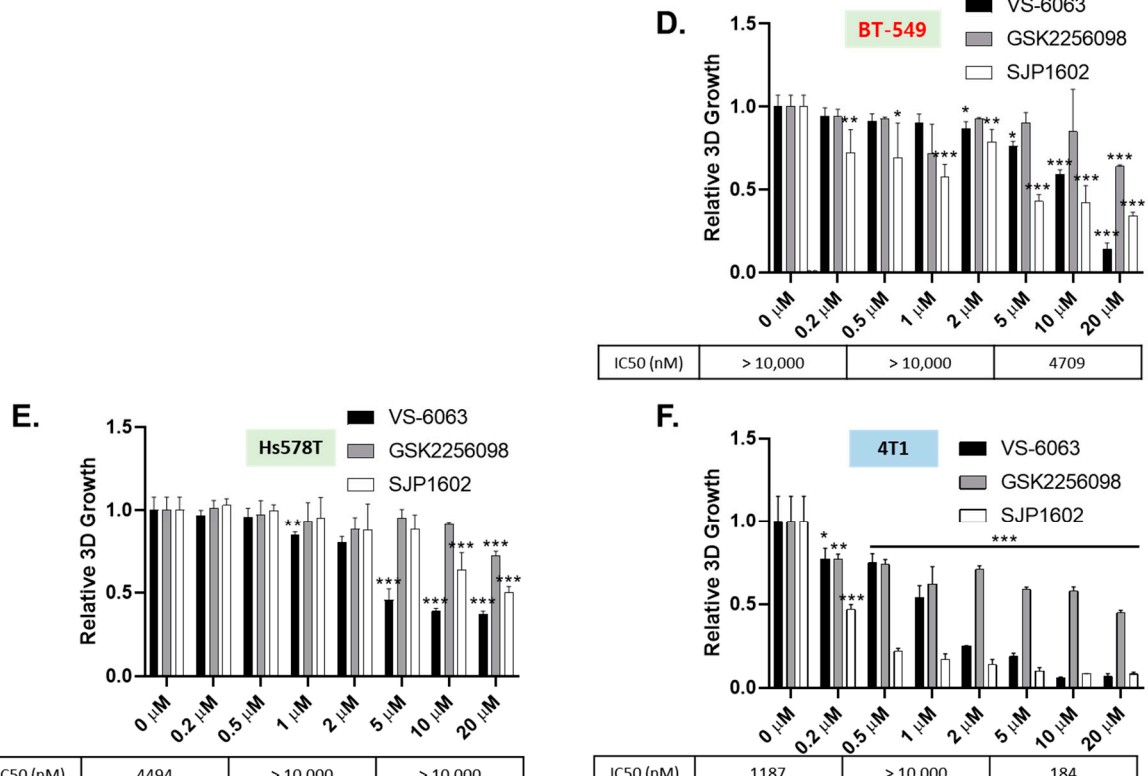

**Figure 3.** Inhibition of 3D single spheroid growth by SJP1602. (**A**) The figure depicts the impact of SJP1602 on the growth of 3D single spheroids formed by MDA-MB-231. Cells were treated with SJP1602 at various concentrations (ranging from 200 nM to 20 μM) for 4 days, following 3-day culture in medium with 2% Matrigel in 96-well ultra-low attachment round plates. Phase contrast microscopy images were captured, and cell viability was quantified for MDA-MB-231 (**A**) and other TNBC cell lines, including HCC70 (**B**), BT-20 (**C**), BT-549 (**D**), Hs578T (**E**), and 4T1 (**F**). Statistical analysis was performed using an unpaired, two-tailed Student's *t*-test, and * $p < 0.05$, ** $p < 0.01$, *** $p < 0.001$ denote significant differences in cell viability compared to the control group.

FAK expression is significantly elevated in invasive human cancers, wherein FAK signaling aids in directional cell movement [11,31,32]. To assess this, the influence of SJP1602 on the migration of MDA-MB-231 cells was investigated via transwell assays. Exposure of cells to 5 μM GSK 2256098 did not affect the migration of MDA-MB-231 cells. However, SJP1602, at the same concentration, considerably reduced serum-stimulated migration (Figure 4A,B). To validate the observations from the migration assay, we performed Boyden chamber motility assays using Matrigel-coated membranes. The incorporation of SJP1602 into the cell invasion assay markedly inhibited the invasion of MDA-MB-231 cells (Figure 4C,D). To further elucidate the inhibitory effect of SJP1602 on the invasion of TNBC cells, we conducted a 3D cell invasion assay using MDA-MB-231, 4T1, and Hs578T cells. MDA-MB-231 cells exhibited substantial invasion into the surrounding matrix when cultured in 3D conditions using Matrigel and Type I Collagen (Figure 5A). At a concentration of 1 μM, SJP1602 significantly inhibits TNBC cell invasion compared to VS-6063 and GSK 2256098 (Figure 5B–D).

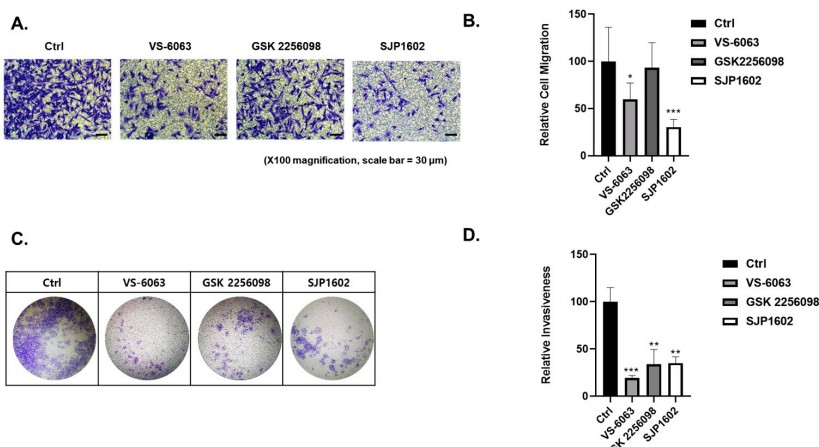

**Figure 4.** Inhibition of cell migration and 2D invasion in MDA-MB-231 by SJP1602. (**A**) Cell migration capacity was analyzed using the transwell migration assay. MDA-MB-231 cells were seeded in the 24-well chamber and treated with vehicle or FAK inhibitors (SJP1602, VS-6063, GSK2256098) at a concentration of 5 μM. After 16 h, migrated cells were analyzed using a microscope. (**B**) Bar diagram illustrates the relative cell migration of MDA-MB-231 cells treated with FAK inhibitors, including SJP1602, VS-6063, and GSK2256098. Statistical analysis was performed using an unpaired, two-tailed Student's *t*-test, and * *p* < 0.05, *** *p* < 0.001 denote significant differences in cell migration compared to the control group. (**C**) Cell invasion capacity was analyzed using the Boyden chamber assay. MDA-MB-231 cells were seeded in the 24-well Boyden chambers with Matrigel-coated filters and treated with vehicle or FAK inhibitors (SJP1602, VS-6063, GSK2256098) at a concentration of 5 μM. After 48 h, cells that invaded the Matrigel were stained using crystal violet and analyzed using a microscope. (**D**) Bar diagram representing the number of invaded cells into the Matrigel, stained with crystal violet. Statistical analysis was conducted using an unpaired, two-tailed Student's *t*-test. ** *p* < 0.01 and *** *p* < 0.001 indicate significant differences in 2D invasion compared to the control group.

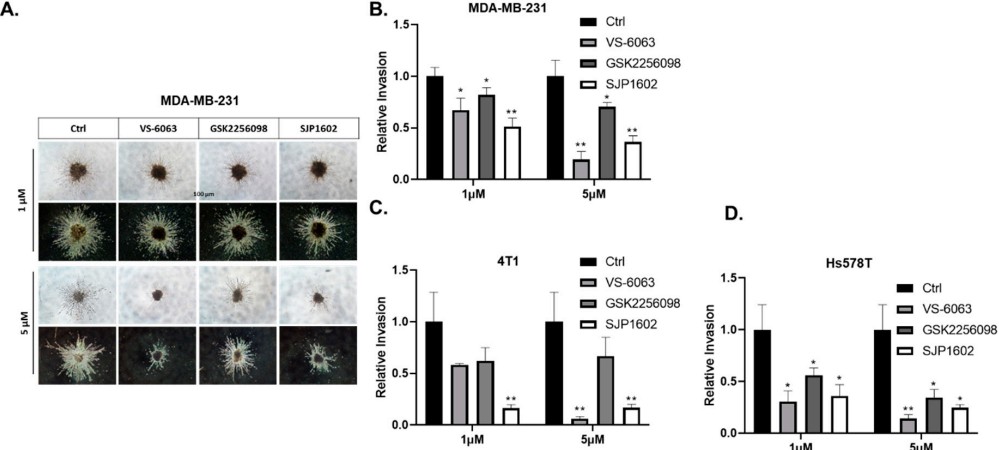

**Figure 5.** Inhibition of 3D invasion of TNBC cells by SJP1602. (**A**) The figure illustrates the impact of SJP1602 on the invasion of 3D single spheroids formed by MDA-MB-231. Cells were added to an ultra-low 96-well round-bottom plate and cultured for 3 days to form spheroids. The medium was removed, and a mixture of neutralized Type I Collagen and Matrigel was added to the wells. After 1 h of incubation, cells were treated with vehicle or FAK inhibitors (SJP1602, VS-6063, GSK2256098) at concentrations of 1 μM or 5 μM. The plate was then incubated for 72 h. Relative invasion was quantified for MDA-MB-231 (**B**), and other TNBC cell lines, including 4T1 (**C**) and Hs578T (**D**). Statistical analysis was performed using an unpaired, two-tailed Student's *t*-test, and * *p* < 0.05, ** *p* < 0.005 denote significant differences in cell invasion compared to the control group.

### 3.4. Pharmacokinetic Studies of SJP1602

The pharmacokinetic properties of SJP1602 were investigated in Sprague Dawley rats, employing both oral (po) and intravenous (iv) delivery methods at doses of 20 mg/kg and 5 mg/kg, respectively (Figure 6A). Blood samples, taken at various intervals up to 24 h post administration, were analyzed to measure SJP1602 concentrations. Post oral administration, the mean peak plasma concentration ($C_{max}$) was determined to be 3274 ng/mL, with a mean time to reach $C_{max}$ ($T_{max}$) of 1 h. Intravenous administration resulted in a $C_{max}$ of 3889 ng/mL and a $T_{max}$ of 0.28 h (Figure 6B). Plasma concentrations derived from a single oral dose of 20 mg/kg suggested high bioavailability (45%) with a half-life of 2.2 h. The area under the curve (AUC) was calculated to be 22 μmol/L, and a peak concentration of 4.9 μmol/L was achieved within 1 h. These data support the ability of SJP1602 to maintain therapeutically relevant plasma concentrations, underscoring its potential efficacy as a treatment option.

**A.**

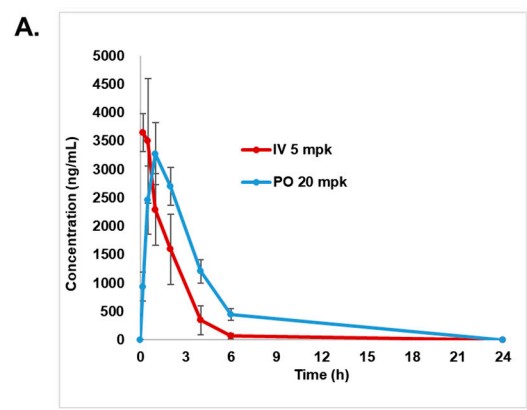

**B.**

| | po (20 mg/kg) | iv (5 mg/kg) |
|---|---|---|
| $AUC_{last}$ (ng·h/mL) | 14,693 ± 2,366 | 7,069 ± 2,505 |
| $C_{max}$(ng/mL) | 3,274 ± 544 | 3,889 ± 751 |
| $T_{max}$ (h) | 1.00 ± 0.00 | 0.28 |
| T1/2 (h) | 2.24 ± 0.19 | 0.93 ± 0.25 |
| Bioavailability (%) | 44.98 | |

**Figure 6.** Pharmacokinetics of SJP1602. (**A**) The plasma concentration–time profile of SJP1602 in rats. SJP1602 was administered orally (po) at a dose of 20 mg/kg and intravenously (iv) at a dose of 5 mg/kg to male SD rats. Blood concentrations of SJP1602 were measured at 0.17, 0.5, 1, 2, 4, 6, and 24 h post administration. (**B**) Pharmacokinetic parameters for SJP1602. The values presented are expressed as the mean ± standard deviation (SD) and the data were obtained from three independent experiments (n = 3). AUC last: area under the curve (AUC) from zero to the last measurable concentration point, $C_{max}$: maximum concentration, $T_{max}$: time to maximum concentration, $T_{1/2}$: half-life.

### 3.5. Antitumor Efficacy of SJP1602 in Xenograft Models of TNBC

To evaluate the inhibitory effect of SJP1602 on primary tumor growth in vivo, we utilized a diverse range of cell transplantation mouse models of TNBC. Treatment with SJP1602 notably inhibited the primary tumor growth of all three tested TNBC cell lines in vivo. In MDA-MB-231 xenograft models, SJP1602 was administered orally in doses of 20, 40, or 80 mg/kg daily, resulting in a notable dose-dependent reduction in tumor growth (Figure 7A). Tumor growth inhibition (TGI), determined on the 14th day, indicated results of 72%, 74%, and 81% ($p < 0.05$) in the cohorts treated with 20, 40, or 80 mg/kg of SJP1602, respectively. Importantly, by the 14th day, a significant reduction was observed in both tumor mass and weight in the group receiving the 80 mg/kg dose of SJP1602 (Figure 7B,C). Similarly, in the BT-549 TNBC model, significant tumor growth inhibition was observed with the administration of SJP1602 at doses of 20 or 80 mg/kg daily. VS-6063 was ineffective at the 80 mg/kg dosage, whereas even 20 mg/kg of SJP1602 showed a pronounced inhibition of cancer growth. Notably, while SJP1602 at 80 mg/kg led to a reduction in tumor mass compared to the control group at day 21, it did not statistically significantly decrease the tumor weight. In the final xenograft model, MDA-MB-453, administration of SJP1602 (either 20 mg/kg or 80 mg/kg) for a duration of 56 days resulted in a substantial growth inhibitory effect, with reductions ranging from 89% to 114% (Figure 7G). It is worth

pointing out that, by the 56th day, both tumor volume and weight had seen a substantial reduction in the group treated with an 80 mg/kg dose of SJP1602 (Figure 7H,I). In the mouse xenograft experiments detailed here, no signs of toxicity or significant weight loss were noted with SJP1602 administration (Supplementary Figure S2). In conclusion, SJP1602 exhibits significant potential in inhibiting tumor growth in diverse mouse xenograft models of TNBC.

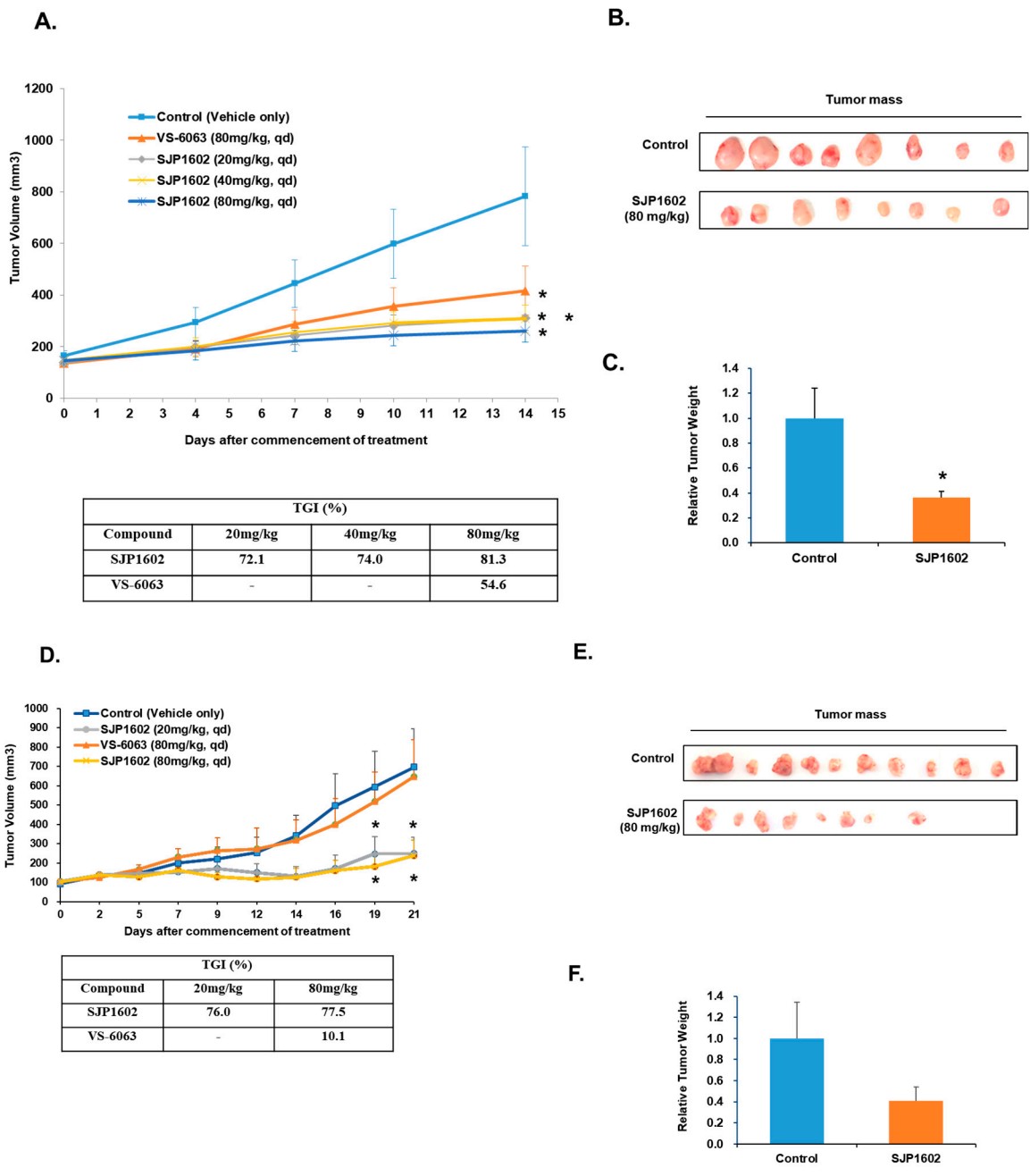

**Figure 7.** *Cont.*

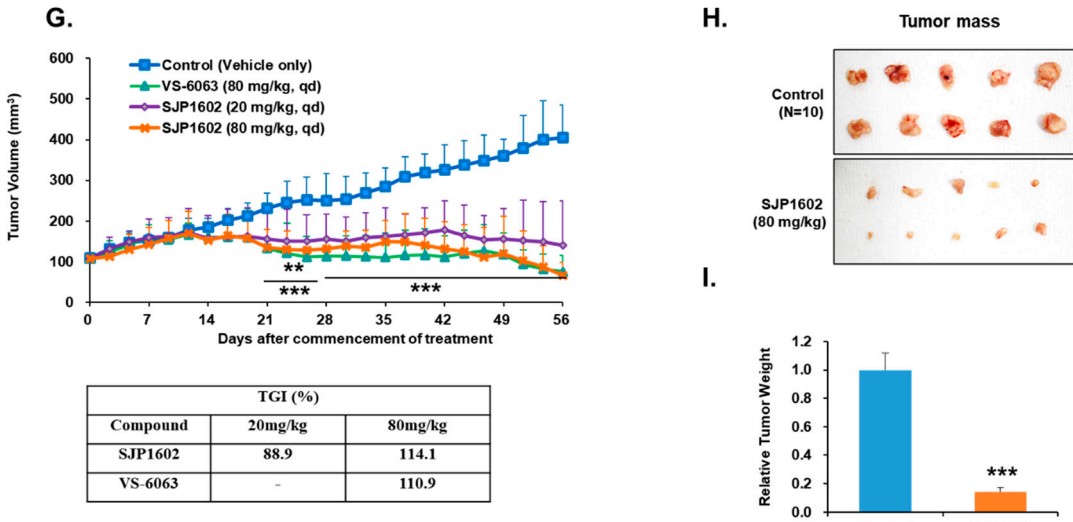

**Figure 7.** Antitumor activity of SJP1602 in xenograft models of TNBC. (**A**) Anti-tumor growth activities of SJP1602 in MDA-MB-231 mouse xenograft models. SJP1602 was orally administered once daily (QD) at doses of 20 to 80 mg/kg. VS-6063 was orally administered once daily (QD) at doses of 80 mg/kg. The error bars represent ± SEM (n = 8). Statistical significance is denoted as * $p < 0.05$ (Student's *t*-test, two-tailed). (**B**) The image of the tumor mass in MDA-MB-231 xenograft tumors. (**C**) Tumor weight of MDA-MB-231 xenograft tumors. Statistical significance is denoted as * $p < 0.05$. (**D**) Anti-tumor growth activities of SJP1602 in BT-549 mouse xenograft models. SJP1602 was orally administered once daily (QD) at doses of 20 to 80 mg/kg. VS-6063 was orally administered once daily (QD) at doses of 80 mg/kg. The error bars represent ± SEM (n = 11). Statistical significance is denoted as * $p < 0.05$ (Student's *t*-test, two-tailed). (**E**) The image of the tumor mass in BT-549 xenograft tumors. (**F**) Tumor weight of BT-549 xenograft tumors. (**G**) Anti-tumor growth activities of SJP1602 in MDA-MB-453 mouse xenograft models. SJP1602 was orally administered once daily (QD) at doses of 20 to 80 mg/kg. VS-6063 was orally administered once daily (QD) at doses of 80 mg/kg. The error bars represent ± SEM (n = 10). Statistical significance is denoted as ** $p < 0.01$ and *** $p < 0.001$ (Student's *t*-test, two-tailed). (**H**) The image of the tumor mass in MDA-MB-453 xenograft tumors. (**I**) Tumor weight of MDA-MB-453 xenograft tumors. *** $p < 0.001$.

## 4. Discussion

TNBC is a subtype of breast cancer known for its aggressive nature and limited targeted treatment options [1,3]. Focal adhesion kinase (FAK) is upregulated in TNBC and plays a pivotal role in promoting tumor growth, invasion, and metastasis [26]. The effect of SJP1602 was observed to effectively inhibit both FAK and PYK2. In vitro studies revealed that SJP1602 outperforms currently used FAK inhibitors such as VS-6063 and exhibits robust efficacy in TNBC cell lines. Specifically, the inhibitory effect of SJP1602 was more pronounced in 3D cultures compared to 2D cultures, highlighting the significance of FAK signaling in a 3D environment where cell survival is more reliant on FAK due to spatial constraints and limited adhesion. The preference for FAK inhibitors in 3D culture arises from the increased reliance of cancer cells on FAK signaling in a 3D environment [29]. In 2D cultures, cells can spread out and adhere to the culture dish surface, providing ample surface area for interaction and survival without relying heavily on FAK signaling. Conversely, in 3D environments, cells are surrounded by other cells and the extracellular matrix, limiting their ability to spread out and increasing their dependence on FAK signaling for survival, migration, and invasion [4].

The effect of SJP1602 extended to TNBC cell growth, migration, and invasion, indicating its potential to impede metastatic progression. In vivo pharmacokinetic studies in rats demonstrated the high bioavailability of SJP1602 and its ability to reach and maintain therapeutically relevant plasma concentrations. Furthermore, the effect of SJP1602 was validated in xenograft models of TNBC, where it led to a significant and dose-dependent

reduction in tumor growth across various TNBC cell lines. A limitation of this study is the absence of mechanism-of-action studies for SJP-1602 in animal efficacy experiments. To address this limitation, we are currently conducting further investigations to determine whether any of the kinases identified in the kinase profiling analysis of SJP-1602 are involved in xenograft efficacy. These additional studies aim to provide insights into the specific molecular pathways and targets through which SJP-1602 exerts its inhibitory effects on tumor growth and metastasis in vivo.

Despite promising results, it is essential to acknowledge that, to date, no FAK inhibitors have achieved clinical success in TNBC treatment. However, the high expression of FAK in TNBC and its association with aggressive tumor characteristics highlight the importance of exploring FAK inhibitors as potential therapeutic options for this specific subtype of breast cancer. Comparatively, SJP1602 and VS-6063 demonstrated superior efficacy in inhibiting proliferation and metastasis in TNBC cells compared to the selective FAK inhibitor GSK 2256098, which does not target PYK2. The dual inhibition of both FAK and PYK2 by SJP1602 contributed to its potent effects, whereas the selective targeting of FAK alone by GSK 2256098 showed limited effectiveness in blocking cancer cell proliferation and metastasis. The higher potency of SJP1602 in inhibiting the kinase activity of both FAK and PYK2 translated to enhanced effectiveness in TNBC cell lines and xenograft models. This highlights the importance of simultaneous inhibition of both FAK and PYK2 for achieving optimal therapeutic outcomes in TNBC treatment. These findings underscore the potential of SJP1602 as a promising candidate for further preclinical and clinical evaluation in the context of TNBC treatment. The potent inhibitory effect of SJP1602 on the kinase activity of both FAK and PYK2 offers promising prospects for advancing targeted therapies against TNBC.

Ongoing investigations are currently assessing the efficacy of SJP1602 in tumor models in conjunction with established drug classes [6]. Moreover, the research is being extended to encompass other types of cancer, such as malignant pleural mesothelioma [33]. Preliminary observations indicate that certain mesothelioma cell lines demonstrate sensitivity to the effects of SJP1602. These findings suggest potential broader applications for SJP1602 in targeting other cancer types beyond TNBC.

**Supplementary Materials:** The following supporting information can be downloaded at: https: //www.mdpi.com/article/10.3390/cimb45090446/s1, Figure S1: Effect of SJP 1602 on TNBC Cells; Figure S2: Changes in Body Weight of TNBC Xenograft Mouse Models Following Treatment with SJP 1602; Table S1: Kinase Assays for SJP1602; Table S2: Cytotoxicity Evaluation of SJP1602 on Normal Cell Lines.

**Author Contributions:** Conceptualization, M.J. and S.H.; methodology, H.C.; software, H.P.; validation, S.-M.L.; formal analysis, M.J.; investigation, S.H.; resources, M.J.; data curation, S.-M.L.; writing—original draft preparation, S.H.; writing—review and editing, S.H.; visualization, M.J.; supervision, S.A.; project administration, S.A.; funding acquisition, S.A. All authors have read and agreed to the published version of the manuscript.

**Funding:** Incheon National University Research Grant in 2019.

**Institutional Review Board Statement:** All animal experiments were carried out in accordance with the ethical guidelines and approved by the Institutional Animal Care and Use Committee (IACUC) at the following institutions: (1) Samjin Pharmaceutical Co., Seoul, Republic of Korea (Approval Code: SJ-2018-004). Approval Period: 1 December 2018–30 November 2019. (2) Daegu-Gyeongbuk Medical Innovation Foundation (DGMIF), Daegu, Republic of Korea (Approval Code: DGMIF-18041702-01). Approval Period: Starting from 20 January 2015. (3) Chaon, Seongnam, Republic of Korea (Approval Code: CE2019094). Approval Period: 3 October 2019–31 January 2020.

**Informed Consent Statement:** Not applicable.

**Data Availability Statement:** The data that support the findings of this study are available on request from the corresponding author Soon Kil Ahn.

**Acknowledgments:** This work was supported by the Incheon National University Research Grant in 2019 and by the National Research Foundation of Korea (RS-2023-00227084).

**Conflicts of Interest:** Authors Myeongjin Jeon, Hyoungmin Cho, and Soo-Min Lee were employed by the company Samjin Pharm. Co., Ltd. The remaining authors declare that the research was conducted in the absence of any commercial or financial relationships that could be construed as a potential conflict of interest.

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
