# Peer review of "Targeting FAK/PYK2 with SJP1602 for Anti-Tumor Activity in Triple-Negative Breast Cancer"

_cimb, doi:10.3390/cimb45090446_

Round 1

Reviewer 1 Report

Triple-negative breast cancer (TNBC) poses significant challenges due to its aggressive nature and limited treatment options. Focal adhesion kinase (FAK) has emerged as a critical factor promoting tumor growth and metastasis in TNBC. This study investigated the role of FAK/PYK2 inhibitor, SJP1602, in antitumor activity in TNBC. The authors demonstrated that SJP1602 inhibited cell growth, migration, invasion, and 3D spheroid formation in TNBC cells in a concentration-dependent manner. Furthermore, in TNBC xenograft models, SJP1602 exhibited significant dose-dependent inhibition of tumor growth. The topic of this study is interested. However, there are some issues to consider.

1.      Please check the cell culture conditions. Most of the TNBC cell lines could not grow well in a humidified atmosphere with 5% CO2, including MDA-MB-231 and MDA-MB-453.

2.      Which commercially available kit was used in FAK [pY397] enzyme-linked immunosorbent assay?

3.      No LC‑MS/MS conditions were described in Materials and Methods section.

4.      No statistical analysis was described in table 1, table 2, figure 2 and figure 4B.

5.      Animal weights during the tumor xenograft study were not shown in the study.

6.      What is the IC50 of these inhibitors in TNBC or normal epithelial cell viability?

Triple-negative breast cancer (TNBC) poses significant challenges due to its aggressive nature and limited treatment options. Focal adhesion kinase (FAK) has emerged as a critical factor promoting tumor growth and metastasis in TNBC. This study investigated the role of FAK/PYK2 inhibitor, SJP1602, in antitumor activity in TNBC. The authors demonstrated that SJP1602 inhibited cell growth, migration, invasion, and 3D spheroid formation in TNBC cells in a concentration-dependent manner. Furthermore, in TNBC xenograft models, SJP1602 exhibited significant dose-dependent inhibition of tumor growth. The topic of this study is interested. However, there are some issues to consider.

1.      Please check the cell culture conditions. Most of the TNBC cell lines could not grow well in a humidified atmosphere with 5% CO2, including MDA-MB-231 and MDA-MB-453.

2.      Which commercially available kit was used in FAK [pY397] enzyme-linked immunosorbent assay?

3.      No LC‑MS/MS conditions were described in Materials and Methods section.

4.      No statistical analysis was described in table 1, table 2, figure 2 and figure 4B.

5.      Animal weights during the tumor xenograft study were not shown in the study.

6.      What is the IC50 of these inhibitors in TNBC or normal epithelial cell viability?

Author Response

1. Please check the cell culture conditions. Most of the TNBC cell lines could not grow well in a humidified atmosphere with 5% CO2, including MDA-MB-231 and MDA-MB-453.

Response 1: We have successfully cultured both MDA-MB-231 and MDA-MB-453 cell lines in a humidified atmosphere with 5% CO2 at 37℃. The culture medium used was RPMI Medium 1640 (RPMI) supplemented with 10% FBS, 25mM HEPES, and penicillin

2. Which commercially available kit was used in FAK [pY397] enzyme-linked immunosorbent assay?

Response 2: For the phospho-FAK ELISA, we followed the manufacturer's instructions using the FAK (Phospho) [pY397] Human ELISA Kit from Thermo Fisher (Catalogue #KHO0441).

3. No LC‑MS/MS conditions were described in Materials and Methods section.

Response 3: According to the reviewer’s suggestion, we corrected the manuscript (Line 241-250).

4. No statistical analysis was described in table 1, table 2, figure 2 and figure 4B.

Response 4: According to the reviewer’s suggestion, we corrected the manuscript (table 1, table 2, figure 2B and figure 4D) and (Line 323-324, 337, 350-352, 406-408)

5. Animal weights during the tumor xenograft study were not shown in the study.

Response 5: According to the reviewer’s suggestion, we corrected the manuscript (Supplementary Figure2) and (Line 462-463)

6. What is the IC50of these inhibitors in TNBC or normal epithelial cell viability?

Response 6: In our tests, SJP1602 showed no cytotoxic effects against the normal cell lines we examined, including CCD-18Co, WI38, Fa2N4, FDF, and HEK293T. This information is detailed in Supplementary Table 2. (Line 344-345)

Reviewer 2 Report

1) 2.5. FAK [pY397] Enzyme-Linked Immunosorbent Assay. Can the author provide a western blot assay to detect phospho-FAK levels

2) Does SJP1602 also bind with other target? how the author exclude this possibility?

3) How about toxicity of SJP1602 in vivo? At least the author should weight change figure in Figure 7. 

Author Response

1) 2.5. FAK [pY397] Enzyme-Linked Immunosorbent Assay. Can the author provide a western blot assay to detect phospho-FAK levels

Response 1: Western blot assays to detect phospho-FAK levels were conducted, and the results are presented in Supplementary Figure 1. For these assays, TNBC cells were treated with SJP1602 at concentrations equivalent to the IC50 for pFAK, and the expression and phosphorylation of FAK proteins were assessed (Line 333-334).

2) Does SJP1602 also bind with other target? how the author exclude this possibility?

Response 2: While it is quite possible that SJP1602 binds to other kinases as well, it was assessed against a panel of kinase assay, as outlined in Supplementary Table 1. At a concentration of 1 μM (270-fold the IC50 for FAK), SJP1602 inhibited MET and ALK by 79% and 72%, respectively. However, for the other 106 kinases, inhibition was less than 50%. Thus, while SJP1602 can bind to other kinases, its inhibitory potency is more pronounced for FAK and PYK2, suggesting selectivity towards these targets.

3) How about toxicity of SJP1602 in vivo? At least the author should weight change figure in Figure 7. 

Response 3: In the mouse xenograft experiments detailed here, no signs of toxicity or significant weight loss were noted with SJP1602 administration (Supplementary Figure2). According to the reviewer’s suggestion, we corrected the manuscript (Supplementary Figure2) and (Line 462-463).

Reviewer 3 Report

Triple-negative breast cancer is known to be a significant problem due to its aggressive nature and limited treatment options. Focal adhesion kinase (FAK) is currently being actively studied as a critical factor in promoting tumor growth and metastasis in TNBC. Several preclinical and early clinical trials of various FAK inhibitors have been performed, but none of them has yet achieved clinical success in the treatment of TNBC, which underlines the relevance of research in this direction.

The authors of the article are investigating the therapeutic potential of a new dual inhibitor of FAK and PYK2, named SJP1602, for TNBC. In vitro experiments show that SJP1602 effectively inhibits both FAK and PYK2 activity, showing potent effects on both kinases. The authors conducted a study both on cell lines and xenografts of breast tumors. SJP1602 shows concentration-dependent inhibition of growth, migration, invasion, and also shows a significant dose-dependent inhibition of tumor growth. These promising results highlight the potential of SJP1602 as a potent dual inhibitor of FAK and PYK2, deserving further investigation in clinical trials for the treatment of TNBC.

The article logically and consistently presents the material, the design of the study does not raise questions. I believe that this material undoubtedly deserves publication and will be of interest to readers.

Small remarks:

1. In fig. 4e does not show variation intervals.

2. Why is the dosage of inhibitors up to 80 mg/kg chosen? how is the upper limit chosen?

Author Response

The article logically and consistently presents the material, the design of the study does not raise questions. I believe that this material undoubtedly deserves publication and will be of interest to readers.
Small remarks:

1. In fig. 4D does not show variation intervals.
Response 1: According to the reviewer’s suggestion, we corrected the manuscript (figure 4D)

2. Why is the dosage of inhibitors up to 80 mg/kg chosen? how is the upper limit chosen?
Response 2: The dosage for the in vivo efficacy test was established with a low concentration of 20mpk and a high concentration of 80mpk. This range was chosen based on the effective dose of VS-6063, a reference FAK inhibitor, to allow for a comparative evaluation with VS-6063.

Round 2

Reviewer 1 Report

The manuscript has been revised and improved. However, there are still some issues to consider before publication.

l   Line 108, line 443, line 504, line 514. “In vivo” should be revised to “in vivo” in italic type.

l   Line 14, line 119, line 315, line 491. “In vitro” should be revised to “in vitro” in italic type.

l   Line 161, line 273. “CO2” should be revised to “CO2”.

l   Line 274, line 331, line 335. “IC50” should be revised to “IC50”.

The manuscript has been revised and improved. However, there are still some issues to consider before publication.

l   Line 108, line 443, line 504, line 514. “In vivo” should be revised to “in vivo” in italic type.

l   Line 14, line 119, line 315, line 491. “In vitro” should be revised to “in vitro” in italic type.

l   Line 161, line 273. “CO2” should be revised to “CO2”.

l   Line 274, line 331, line 335. “IC50” should be revised to “IC50”.

Author Response

(1)   Line 108, line 443, line 504, line 514. “In vivo” should be revised to “in vivo” in italic type.

Response 1: According to the reviewer’s suggestion, we corrected the manuscript (Line 108, line 443, line 504, line 514)

(2)   Line 14, line 119, line 315, line 491. “In vitro” should be revised to “in vitro” in italic type.

Response 2: According to the reviewer’s suggestion, we corrected the manuscript (Line 14, line 119, line 315, line 491)

(3)   Line 161, line 273. “CO2” should be revised to “CO2”.

Response 3: According to the reviewer’s suggestion, we corrected the manuscript (Line 161, line 273)

(4)  Line 274, line 331, line 335. “IC50” should be revised to “IC50”.

Response 4: According to the reviewer’s suggestion, we corrected the manuscript (Line 274, line 331, line 335)